# Autocatalytic Tissue Polymerization Reaction Mechanism in Colorectal Cancer Development and Growth

**DOI:** 10.3390/cancers12020460

**Published:** 2020-02-17

**Authors:** Bruce M. Boman, Arthur Guetter, Ryan M. Boman, Olaf A. Runquist

**Affiliations:** 1Center for Translational Cancer Research, Helen F. Graham Cancer Center & Research Institute, Newark, DE 19718, USA; brboman@udel.edu; 2Department of Biological Sciences, University of Delaware, Newark, DE 19716, USA; 3Department of Pharmacology & Experimental Therapeutics, Thomas Jefferson University, Philadelphia, PA 19107, USA; 4Department of Mathematics, Hamline University, Saint Paul, MN 55104, USA; aguetter@hamline.edu; 5Department of Materials Science & Engineering, Drexel University, Philadelphia, PA 19104, USA; 6CATX, Inc., Princeton, NJ 08542, USA; 7Department of Chemistry, Hamline University, Saint Paul, MN 55104, USA; orunquist@hamline.edu

**Keywords:** colon cancer, autocatalytic reaction, tissue polymerization, logistic model, nonlinear function, cancer stem cells, *APC* gene, familial adenomatous polyposis, crypt fission, tissue renewal

## Abstract

The goal of our study was to measure the kinetics of human colorectal cancer (CRC) development in order to identify aberrant mechanisms in tissue dynamics and processes that contribute to colon tumorigenesis. The kinetics of tumor development were investigated using age-at-tumor diagnosis (adenomas and CRCs) of familial adenomatous coli (FAP) patients and sporadic CRC patients. Plots of age-at-tumor diagnosis data as a function of age showed a distinct sigmoidal-shaped curve that is characteristic of an autocatalytic reaction. Consequently, we performed logistics function analysis and found an excellent fit (*p* < 0.05) of the logistic equation to the curves for age-at-tumor diagnoses. These findings indicate that the tissue mechanism that becomes altered in CRC development and growth involves an autocatalytic reaction. We conjecture that colonic epithelium normally functions as a polymer of cells which dynamically maintains itself in a steady state through an autocatalytic polymerization mechanism. Further, in FAP and sporadic CRC patients, mutation in the adenomatous polyposis coli (*APC*) gene increases autocatalytic tissue polymerization and induces tumor tissues to autocatalyze their own progressive growth, which drives tumor development in the colon.

## 1. Introduction

Colorectal cancer (CRC) is the second leading cause of cancer deaths, with a mortality rate of ~50,000 deaths/year in the US and ~700,000 deaths/year worldwide [1,2]. The scientific community now widely accepts that CRC develops through an adenoma carcinoma sequence [3,4] and that mutations in the adenomatous polyposis coli (*APC*) gene are key driver mutational events in the the initiation and development of most (>80%) CRCs [5]. *APC* mutations are acquired in the development of CRC in sporadic cases and inherited in hereditary CRC cases that occur in the familial adenomatous polyposis coli (FAP) syndrome. Indeed, investigations of FAP led to the identification, mapping and isolation of the *APC* gene. FAP is an autosomal dominant trait caused by inheritance of a germline *APC* mutation. Classic FAP patients develop 100s to 1000s of premalignant adenomas which further supports the idea that *APC* mutations drive tumor growth in vivo. If FAP patients are left untreated, they have nearly a 100% risk of developing CRC. Adenomas in FAP patients show loss of the 2nd wild-type *APC* allele [6,7]. Three hits at *APC* have also been shown to occur in tumors from attenuated FAP (AFAP) patients who have a milder phenotype with fewer colon polyps (average of 30) and a later age of colon cancer development [8,9,10]. Two hits at the *APC* locus also occur as acquired mutations in the development of most sporadic CRCs [11]. Additionally, a comprehensive screening of CRC cell lines and primary CRCs revealed that third hits at APC are acquired in some sporadic tumors [12]. In both FAP and sporadic cases, additional genetic alterations are also acquired during the evolution of advanced CRC as commonly defined by the “Vogelgram” [13]. While this sequence of genetic and pathological events in CRC development is well characterized, the kinetics of CRC initiation and progression have not been fully elucidated. 

To investigate the kinetics underlying the CRC development in humans, we studied age-at-tumor diagnosis (pre-cancerous adenomas and CRC) in both FAP and sporadic patients. While FAP is relatively uncommon (incidence = 1.90 × 10^−6^; prevalence = 4.65 × 10^−5^) [14]), results reported here should have wider implications for understanding mechanisms involved in the development of commonly occurring sporadic adenomatous polyps (1 in 2 individuals) and sporadic CRC (1 in 20 individuals) [15]. As noted above, in both FAP and sporadic cases, mutations of *APC* genes are known events in the development of colonic tumors [6,7,8,9,10,11,12]. Study of the in vivo rate of somatic *APC* mutation also supports the theory that FAP and sporadic CRC follow the same genetic pathway [16]. Because a critical part of identification of tumorigenic mechanisms requires testing proposed processes with quantitative data, a useful mechanistic process for CRC development must provide logical links between quantitative data and qualitative histopathologic information. Our goal here is to quantitatively determine the kinetics of adenoma and CRC development in FAP and sporadic cases in order to identify mechanisms that may explain how changes in tissue dynamics and processes contribute to CRC development. We selected hereditary and sporadic CRC to investigate because CRC development progresses in both cases along the lines of normal colon to pre-cancerous colonic adenomas to CRC, and because data were available from various population studies on FAP and sporadic patients [17,18,19].

## 2. Methods

In our study, we obtained age-at-adenoma and age-at-CRC data from the literature and publicly available databases [17,18,19]. The age-at-adenoma diagnosis (*n* = 293 cases) and age-at-CRC diagnosis (*n* = 214 cases) for FAP patients was obtained from the propositus cohort (not the call-up cohort) in the St Marks Registry and they all had the classical FAP phenotype (100–1000+ polyps) [17]. Classic FAP patients are known to have germline mutations in *APC*’s middle region, particularly at codon 1309 [6,7]. The age-at-adenoma diagnosis for sporadic patients (*n* = 1860) was obtained from the National Polyp Study [18]. The age-at-colon cancer diagnosis for sporadic CRC patients in the US was obtained from The Surveillance, Epidemiology, and End Results (SEER) cancer incidence database [19].

To measure kinetics involved in CRC development, we plotted age-at-tumor diagnosis in FAP patients and sporadic patients as a function of age (Figure 1 and Figure 2). These plots represent the age distribution for the cohort of patients who develop CRC and the data are extrapolated and normalized to denote the fraction (0.0 to 1.0) of tumor patients in this cohort based on the range of age-at-tumor diagnoses as reported in the literature. Inspection of these graphs showed that they are clearly sigmoidal S-shaped curves. Based on this distinct sigmoidal shape and that S-shaped curves are typical of autocatalytic reactions [20,21], we surmised that the dynamics of CRC development could involve an autocatalytic reaction. We do note that S-shaped curves are not proof of an autocatalytic reaction because they are also characteristic of other processes including growth of populations. Notably, sigmoid shaped curves also characterize autocatalytic polymerization reactions [22,23] and colonic epithelia can be viewed as a polymer of colonocytes. Moreover, autocatalysis is a mechanism that describes self-replication [23,24] and colonic epithelium is a self-replicating system that occurs through crypt renewal and crypt fission [25,26,27,28]. Thus, we elected to further investigate the possibility of a role of an autocatalytic mechanism in colonic tumor formation.

Because the graphs for age-at-tumor diagnosis data are sigmoid curves (Figure 1 and Figure 2), which are typical for autocatalytic reactions, we performed fitting of a logistic equation to the data using an equation for logistic functions [29,30]. Logistic functions are good models of autocatalytic reactions. Analysis of fit of a logistic curve to the data was based on the following expression:$$y(t)=\frac{\gamma }{1+\alpha e^{-kt}}$$

The model has three parameters: γ = 1 (maximum Y-axis value), α = an arbitrary constant, and *k* = logistic growth rate or steepness of the curve (slope of rising part of the curve). If *k* is positive, the logistic function will increase; if *k* is negative, the function will decrease.

Curve fitting and statistical analyses were done using Microsoft Excel 2013, Microsoft Office, Mathematica.exe, Wolfram Research, Inc. and GraphPad Software, Inc., San Diego, CA, USA.

## 3. Results

Inspection of all the graphs for age-at-tumor diagnosis data (Figure 1 and Figure 2) shows that they are typical sigmoid-shaped autocatalytic-type reaction curves. All of the graphs have a very similar S-shaped curve. Both curves for age-at-tumor diagnosis in sporadic CRC patient populations are right shifted compared to the curves for FAP patient populations. The half-life (*t*_1/2_) or inflection points of age-at-tumor diagnosis plots for FAP adenomas is 17.5 years of age; for FAP CRCs, it is 34 years; for sporadic adenomas, it is 62 years; for sporadic CRCs, it is 68 years. The logistic function analysis shows an excellent fit (*p* < 0.05) of the logistic equation to the graphs of age-at-tumor diagnosis (Figure 3, Figure 4, Figure 5 and Figure 6, Table 1). While this analysis does not provide proof that the mechanism is autocatalytic, it does provide compelling evidence for an autocatalytic reaction. Indeed, the S-shape curves of the logistic function indicate that the growth process proceeds slowly at the start followed by exponential growth and then by a state in which growth slows and levels off approaching a maximum upper limit.

## 4. Discussion

The main message from this study is that our analysis of the age-at-tumor diagnosis for adenomas and CRCs in FAP and sporadic patients suggests that the mechanism for colon cancer development and growth involves an autocatalytic polymerization reaction. The similar shape of the curves indicates that the mechanism for tumor growth is the same in FAP and sporadic cases, which is consistent with studies that show that the same types of gene mutations and pathologic changes occur during the adenoma to CRC development in FAP and sporadic patients. The right shift in the curves for sporadic cases compared to FAP cases is explained by the fact that in most sporadic tumors, a greater number of acquired *APC* mutations occur in sporadic CRCs compared to FAP tumors because FAP patients have an inherited germline *APC* mutation. The increased value of the *k* rate constant for FAP adenomas (Figure 3, Table 1) suggests that the rate of adenoma growth is accelerated in FAP compared to sporadic adenomas, which is consistent with the fact that FAP patients develop multiple polyposis at an early age.

While S-shaped curves are not proof of an autocatalytic mechanism, the goodness of fit with the logistic function does provide compelling evidence for an autocatalytic reaction. Indeed, the logistic equation has three parameters so that comparison of the curves involves three characteristics, not just one or two measures. Specifically, sigmoid shaped curves can also characterize autocatalytic polymerization reactions [22,23], which agrees with biology if colonic epithelia are viewed as a polymer of colonocytes. Moreover, autocatalysis is a mechanism that describes self-replication [23,24] and colonic epithelium is a self-replicating system that occurs through crypt renewal and crypt fission [25,26,27,28]. Further, both crypt renewal and crypt fission rates are increased in FAP epithelium [26,28,31,32,33,34,35]. Assuming that an autocatalytic polymerization reaction contributes to colonic tumor formation, the fact that the age-at-adenoma diagnosis curves are sigmoid shaped suggests that this putative autocatalytic reaction occurs early in adenoma morphogenesis. Since the sporadic adenoma curve is right shifted compared to the FAP adenoma curve and both are S-shaped, this finding suggests that at least two hits in *APC* are required for autocatalytic polymerization to increase in colon tumor formation. Given that these findings suggest the possibility of an autocatalytic mechanism in colonic tumor development and growth, several points warrant further discussion.

### 4.1. What Is Known About Autocatalysis and Autocatalytic Polymerization Reactions from Chemistry and Biology?

Many examples of autocatalytic mechanisms can be found in chemistry and biology. In these scientific fields, a reaction is said to be autocatalytic if one of the reaction products is also a catalyst for the same or a coupled reaction. In other words, the reaction product catalyzes its own formation. The key feature of the corresponding rate equations for autocatalysis is that they describe nonlinear second order reactions. Examples of autocatalytic reactions in chemistry are found in acid-catalyzed hydrolyses of esters and other compounds [20]. The classic examples of autocatalysis in biology are enzymatic reactions and metabolic cycles in glycolysis, particularly phosphofructokinase (PFK) [21]. Such catalytic mechanisms lead to a lowering of the energy of activation of the reaction. Autocatalysis also occurs in polymerization reaction mechanisms in chemistry and biology.

In some polymer reactions, a polymer can catalyze its own polymerization. For example, in cell biology, the autocatalytic polymerization kinetics of the cytoskeletal actin network provide a basic mechanism that explains the persistent random walk of a crawling cell [36]. In tissue biology, the polymerization-like kinetics of branching morphogenesis provides a basic mechanism that explains how complex branched epithelial structures in mammalian tissues develop as a self-organized process [37]. A polymerization reaction involves a process of monomer units reacting together in a chemical reaction to form polymer chains or three-dimensional networks [38]. Polymerization reactions involve three dynamic processes, initiation, polymer growth and termination of the polymer. Thus, polymerization reactions involve depolymerization as well as polymerization processes such as the dynamics involved in growing and shrinking of microtubules. If we consider colon tissue to be a polymer of cells, then cells would be the monomers that adhere together to form the tissue polymer and the extrusion or apoptosis of cells would be the depolymerization process. In this view, tissue renewal/maintenance could involve a balance between polymerization and depolymerization of cells and, in this situation, an autocatalytic reaction mechanism contributes to colonic tissue by catalyzing its own polymerization. If a putative autocatalytic tissue polymerization reaction becomes increased, this altered state could theoretically contribute to progressive tissue growth and tumor development.

### 4.2. How Might Gene Mutations that Occur in Colonic Tumors Be Linked to an Autocatalytic Polymerization Reaction?

As noted above, mutations in the *APC* gene that occur in the development of CRC are drivers of tumor growth. In FAP patients, all the cells in their colon will have an *APC* mutation at birth. The acquisition of an additional *APC* mutation(s) appears necessary for adenoma development so the occurrence of the additional *APC* alteration is likely a rate limiting step in CRC development. Recent studies on the big bang model of human colorectal tumor development indicate that tumor tissue expansion occurs after the initial mutational events in *APC*. The big bang model was postulated by Sottoriva et al. [39] based on a study that performed genomic profiling of individual glands from different regions within a CRC. This genotyping showed (i) a lack of selective sweeps of genetic changes, (ii) uniformly high intratumoral heterogeneity and (iii) subclone mixing in the distant regions of the tumor. The findings revealed that tumors grow predominantly as a single expansion that produces numerous intermixed subclones that are not subject to stringent selection. In their model, it is surmised that both public (clonal) and most detectable private (subclonal) alterations arise early during tumor growth. The profiling of subclones revealed that they wholly retained the original mutation in *APC*, which indicates the *APC* mutation is one of the earliest initiating events that triggers colon tumorigenesis. 

Our results suggesting that an autocatalytic tissue polymerization reaction mechanism contributes to CRC formation may help clarify how, according to the big bang model, early tumor expansion happens and produces numerous intermixed subclones. For example, initial inactivation of *APC* could increase autocatalytic polymerization of tumor tissues that induces tumors to grow predominantly as a single expansion. Further, through a process termed epistasis [40], an initial inactivation of *APC* might produce numerous intermixed subclones by sensitizing the tumor cell to the proliferative effects of acquired passenger-type mutations in cells that would otherwise not be affected by these mutations. In this way interaction between driver and passenger mutations could have non-linear effects leading to increased proliferation in tumor subclones Thus, theoretically only when an *APC* driver mutation occurs in a cell will a passenger mutation have a deleterious effect and the subclone cell populations will then start to grow rapidly.

Another process that might have a similar effect is the emergence of chromosomal instability in tumor cells. Indeed, it is known that *APC* mutations lead to chromosomal instability early in CRC development which appears to contribute to clonal expansion during evolution of CRC [41]. Thus, initial inactivating *APC* events in the colon might lead to an autocatalytic type reaction, epistasis, and genetic instability that might explain early tumor expansion and emergence of various tumor subclones as described in the big bang model of CRC development.

### 4.3. How Might Tissue Changes that Occur in Colonic Tumors Be Tied to an Autocatalytic Polymerization Reaction?

Colonic tissue changes that lead to adenomas usually only involve the development of 1–3 adenomas in sporadic cases whereas in FAP cases a patient develops hundreds to thousands of adenomas. Thus, the formation of a large mass of abnormal crypts, i.e., an adenomatous polyp, is accelerated in FAP compared to sporadic cases. The transformation into adenomas is attributed to an increased rate of crypt fission which is the biologic mechanism involved in adenoma morphogenesis [28,32,42]. 

Crypt fission is a process that normally maintains colonic epithelial homeostasis by generating two identical crypts from one crypt. During human development, as the colon grows in size, crypt fission is responsible for growth of the epithelium [43]. In adults, the rate of crypt fission equals the rate of crypt loss in replacement of crypts to maintain a dynamic balance through a process called the crypt cycle, which is normally controlled by APC [44,45]. However, the rate of crypt fission is increased in FAP patients due to *APC* mutation [24,38,41,42,43,44]. Thus, an increased rate of crypt fission appears to be the main cellular mechanism responsible for adenomatous polyp formation and growth. This mechanism involved in crypt fission appears to resemble an autocatalytic polymerization mechanism because crypt fission is a process that generates more of the same species—i.e., more crypts. Moreover, crypts are made up of sheets in interconnected epithelial cells that can be thought of as polymers of cells. So, crypt fission might be envisioned as a process of polymer branching—a process that is often found in chemical polymer reactions. Indeed, autocatalysis is a mechanism that describes self-replication [23,24] and colonic epithelium is a self-replicating system that occurs via crypt renewal and crypt fission [25,26,27,28]. 

In this view, *APC* mutation might lead to autocatalytic expansion of crypts and produce a large mass of crypts, a.k.a. adenoma morphogenesis. Notably, the rate of crypt fission is increased in FAP patients due to *APC* mutation [28,42,45,46,47,48]. An increased rate of crypt fission appears to be the main cellular mechanism responsible for adenomatous polyp formation and growth. Thus, biological data on crypt fission theoretically support the concept of an autocatalytic tissue polymerization mechanism.

### 4.4. How Might an Autocatalytic Polymerization Mechanism Help Understand the Stem Cell (SC) Origin of Colon Cancer?

Currently, the development of CRC is attributed to “overpopulation of cancer stem cells” and expansion of the so-called “stem cell compartment” [33,34,35,42,49,50,51,52,53,54]. We have shown that *APC* mutations that drive CRC growth do so by causing SC overpopulation [49]. For example, we discovered that ALDH1 identifies SCs in normal and malignant colonic tissues and tracks SC overpopulation during tumor growth in FAP colonic tissues [49]. While *APC* mutations appear to cause colonic SC overpopulation and CRC in humans, the cellular mechanism that leads to SC overpopulation is unclear. Since *APC* mutations lead to an increase in rate of crypt fissioning and fission of one crypt leads to two crypts, this process would increase the number of crypt SCs. In this view, crypt fission is tantamount to symmetric SC division. Consequently, we have proposed that symmetric SC division should be targeted in future therapeutic approaches in oncology [55,56]. Perhaps an understanding of crypt fission as an autocatalytic polymerization process might lead to new treatment approaches. Based on the interpretation of the findings of our study in relation to the biology of CRC, it is conjectured that an increased rate of autocatalytic tissue polymerization contributes to the cancer SC overpopulation that drives development and growth of CRC.

## 5. Conclusions

To identify aberrant mechanisms in tissue dynamics and processes that contribute to colon tumorigenesis, we investigated age-at-tumor diagnosis (adenomas and CRCs) of familial adenomatous coli (FAP) patients and sporadic CRC patients. The plots of age-at-tumor diagnosis as a function of age showed a distinct sigmoidal-shaped curve that characterizes an autocatalytic reaction. Logistics function analysis showed an excellent fit (*p* < 0.05) of the logistic equation to the graphs of age-at-tumor diagnoses. Based on these results, it appears that CRC development occurs through an autocatalytic tissue polymerization reaction mechanism in both FAP and sporadic tumors. We conjecture that colonic epithelium normally functions as a polymer of cells which dynamically maintains tissue homeostasis through an autocatalytic polymerization mechanism. Further, in FAP and sporadic CRC patients, *APC* mutations increase autocatalytic tissue polymerization and induce tumor tissues to catalyze their own progressive growth early in colonic tumor development. Future research into the possibility of this mechanism may help explain why CRC incidence in the elderly (85+) is decreased [57] and why the incidence of CRC is increasing in younger (<50) populations [58].

## 6. Future Directions

Because the autocatalytic tissue polymerization concept presented in this study is still theoretical, other approaches in future studies are needed. For example, such studies might involve the number of adenomas developed at different ages and the natural development of polyps/adenomas in one FAP patient. Moreover, it would be valuable to study other FAP patient cohorts that relate phenotype to the location of the germline mutation within the *APC* gene [59]. Other FAP categories include intermediate and attenuated FAP phenotypes (AFAP). Intermediate FAS patients have a lower polyp phenotype than classic FAP and most have germline mutations in codon 157–1595, excluding the mutation cluster region. AFAP patients have an even milder phenotype (<100 polyps) and later onset of both adenomas and CRC (typically 12 years later compared to classic FAP). Germline mutations in AFAP patients are in the 5′ part, alternative spliced region in exon 9 or in the extreme 3′ site of *APC*.

Other hereditary colon cancer syndromes would also be interesting to study such as ones in which germline mutations lead to an accelerated adenoma–carcinoma progression. Indeed, syndromes in which germline mutation affects DNA repair, such as Lynch syndrome and MUTH-associated polyposis, have accelerated tumor development [60,61]. Tumors in Lynch Syndrome have genetic instability due to microsatellite instability. This also has relevance to sporadic tumors because most CRCs are genetically unstable and the instability exists due to two distinct mechanisms—microsatellite instability (MIN) or chromosomal instability (CIN) [41,62]. MIN exists in a small subset (~15%) of sporadic CRCs due to aberrant mismatch repair. MIN occurs at the nucleotide level, resulting in base substitutions and deletions or insertions of a few nucleotides. In most (85%) other sporadic CRCs, CIN is observed, resulting in gross chromosomal abnormalities such as aneuploidy and loss of heterozygosity (LOH). *APC* mutations are the most common initial molecular alterations in the CIN phenotype [41,62]. Future studies on these various CRC populations with different genetic instabilities may lead to new insights into how autocatalytic tissue polymerization mechanisms might explain tumor pathogenesis.

## Figures and Tables

**Figure 1 cancers-12-00460-f001:**
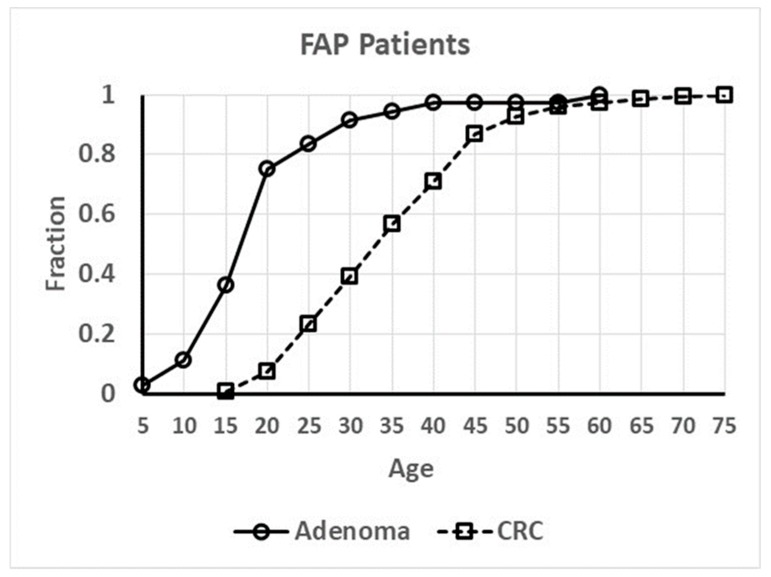
Age at colonic tumor development in familial adenomatous coli (FAP) patients.

**Figure 2 cancers-12-00460-f002:**
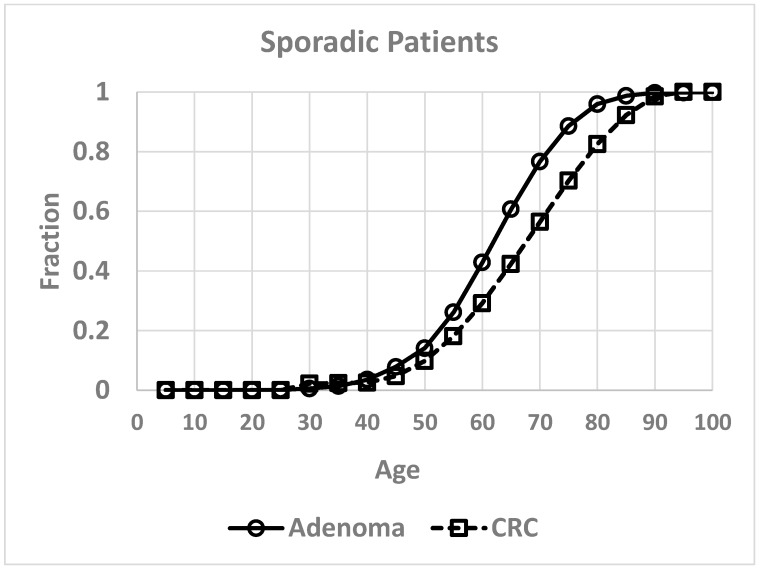
Age at colonic tumor development in sporadic tumor patients.

**Figure 3 cancers-12-00460-f003:**
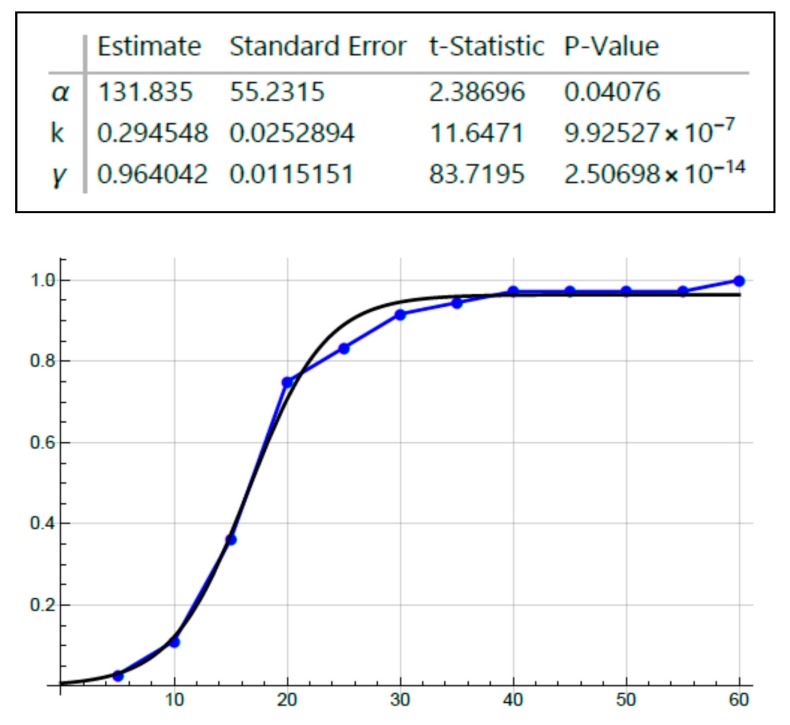
Fitting age-at-adenoma diagnosis data on FAP patients using the logistic equation.

**Figure 4 cancers-12-00460-f004:**
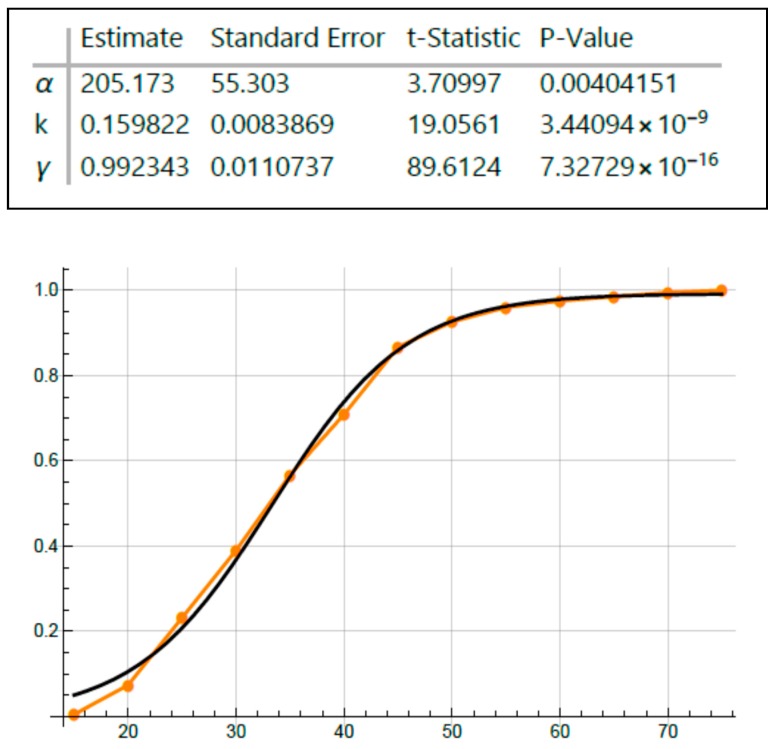
Fitting age-at-adenoma diagnosis data on sporadic tumor patients using the logistic equation.

**Figure 5 cancers-12-00460-f005:**
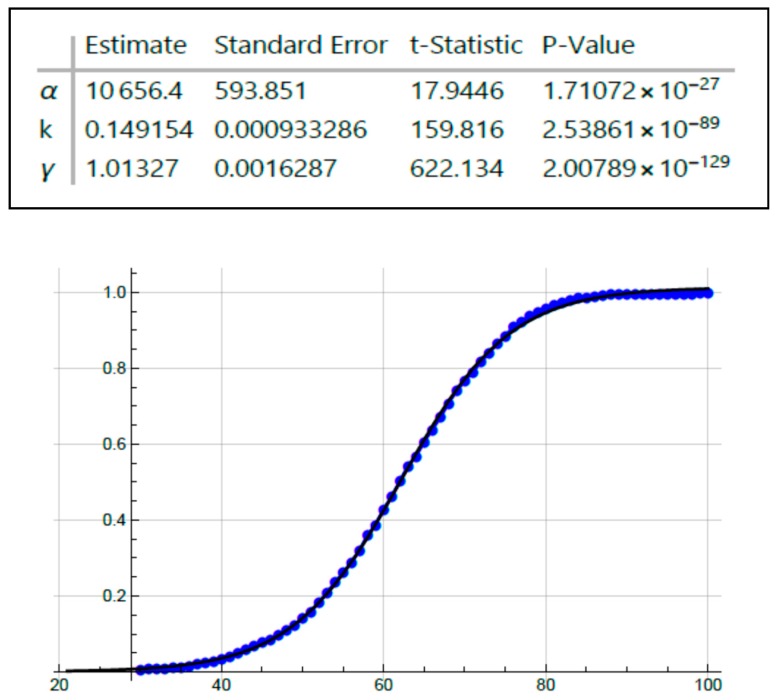
Fitting age-at-colorectal cancer (CRC) diagnosis data on FAP patients using the logistic equation.

**Figure 6 cancers-12-00460-f006:**
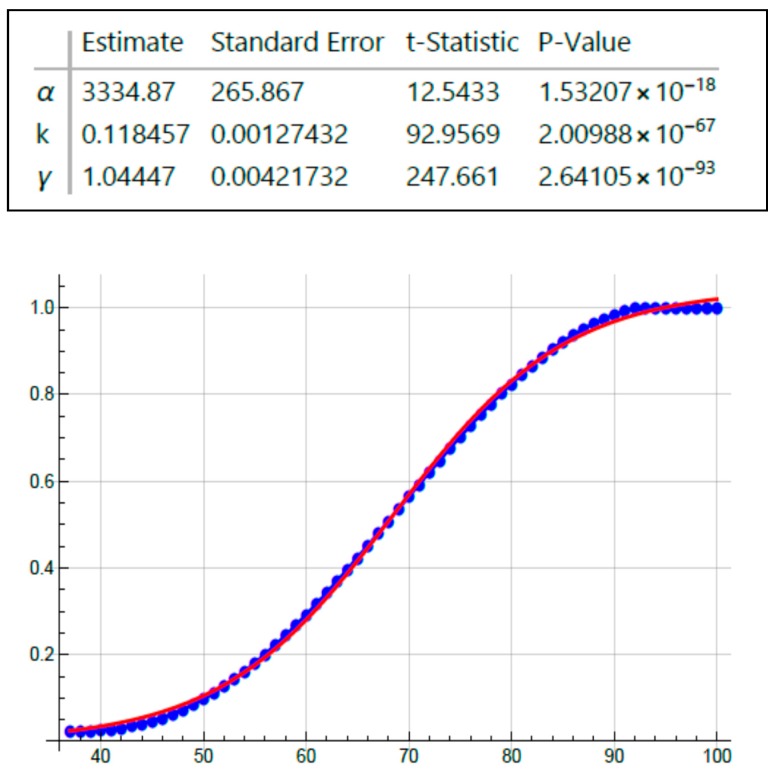
Fitting age-at-CRC diagnosis data on sporadic tumor patients using the logistic equation.

**Table 1 cancers-12-00460-t001:** Summary of the logistic function results.

Tumor	α Value	*k* Value	Inflection Point (Age)
FAP Adenoma	132	0.29	17.5
FAP CRC	205	0.16	34
Sporadic Adenoma	10656	0.15	62
Sporadic CRC	3335	0.11	68

FAP = familial adenomatous polyposis; CRC = colorectal cancer.

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
