# Peer review of "Autocatalytic Tissue Polymerization Reaction Mechanism in Colorectal Cancer Development and Growth"

_cancers, 2020, doi:10.3390/cancers12020460_

Round 1

Reviewer 1 Report

This is an interesting paper, presenting a novel (at least to me!) take on colorectal carcinogenesis. I have two minor queries:

Were all the sporadic cancers studies microsatellite stable 'classical pathway cancers'? Given the evidence for differing biology and faster progression in mismatch repair deficient tumours, these should have been excluded. There is marked genotype-phenotype correlation with respect to colorectal adenoma formation/progression in FAP. There is no mention of the nature of the constitutional pathogenic variant in the FAP patients studied.

'rate of adenoma growth in FAP is greater than sporadic ' - I am to sure what the authors mean by this, or what evidence they have to support it - adenomas do indeed appear early in FAP, but once present most progress very slowly indeed.

Author Response

Responses to Reviewers Comments

We thank the reviewers for their helpful and insightful comments, which we believe have significantly improved our manuscript. Our responses to each comment is followed by the arrowhead (â–º).

Reviewer #1
Comment 1. Were all the sporadic cancers studies microsatellite stable 'classical pathway cancers'? Given the evidence for differing biology and faster progression in mismatch repair deficient tumours, these should have been excluded.

â–ºThe sporadic cancers we analyzed included all colon cancers in the US from SEER database which would mostly include microsatellite stable tumors and there is no way to exclude cancers with microsatellite instability. However, in a separate study, we did normalize to exclude the small proportion of cancers (10%) predicted to have microsatellite instability, it turned out that the age-at-diagnosis curve was still sigmoid shaped that characterizes autocatalytic polymerization reactions.

Comment 2 There is marked genotype-phenotype correlation with respect to colorectal adenoma formation/progression in FAP. There is no mention of the nature of the constitutional pathogenic variant in the FAP patients studied.

â–ºWe apologize for this oversight and now include information that the cohort of FAP patients for age-at-diagnosis of FAP (n = 293) and CRC (n=214) was from the propositus group in the St Marks Registry and they had classical FAP phenotype (100-1000+ polyps). We also now mention that classical FAP patients are known to have germline mutations scattered throughout the amino and middle portions of APC.

Reviewer 2 Report

Dear editor,

This is an interesting new approach investigating tumor development in the colon.

The authors aim to quantitatively determine the kinetics of adenoma and CRC development in FAP and sporadic cases in order to identify mechanisms that may explain how changes in tissue dynamics and processes  contribute to CRC development.

The question is whether authors have used they right methods and the right data to be able to answer this question in my opinion.

I have some mayor issues which I think might undermine the outcomes of this study.

General mayor comments

I really have a problem with the assumption that age at tumor diagnosis can be used as a model for tumor development/progression overall. The age of diagnosis is highly depending on several factors, among which change. Would it not be better to look at the number of adenomas developed at different ages? Or the natural development of polyps/adenomas in one FAP patient?

Also assuming that adenoma /CRC incidence at older age slows down (as expected in an autocatalytic reaction) is problematic in my opinion. The question is also whether polyp development ever stops in a FAP patient, as is suggested by authors. I guess we do not know, it seems that even at an old age people keep developing adenomas.

The less higher incidence of CRC and adenomas at older ages as reported/shown in the graph might be influenced by the fact that in many cases patient have underwent colectomy (in FAP patients) thereby preventing development of CRC . For sporadic cases CRC is being diagnosed less possible not because the incidence goes down but because they are less likely to undergo diagnostic/ colonoscopy because of physical limitations/other diseases.

Also, a problem with using the ‘age-at tumor’ diagnosis in FAP is that it is highly influenced by genotype and family history and whether or not people underwent screening already:

Genotype:

Differences in phenotype highly relate to the location of the mutation within the APC gene.

Three FAP patient groups can be separated: severe/classic , moderate/intermediate and mild/attenuated FAP patients. Classic FAP patients have already developed numerous adenomatous polyps in their colon in their twenties making a colectomy at a relative young age necessary. These patients usually have mutations occurring between codon 1250 and 1464 but particularly at codon 1309. A somewhat less severe, intermediate phenotype (i.e. 100 to thousands of polyps) is associated with mutations in codon 157–1595, excluding the mutation cluster region. Patients with attenuated FAP (AFAP) have an even milder phenotype with less than a hundred adenomas and a later onset of both adenomas and CRC. CRC will develop 12 years later compared with classic FAP. Most of these patients have a mutation in the 5' part, alternative spliced region in exon 9 or in the extreme 3' site of the APC gene. (text from Nielsen et al , chapter on FAP in hereditary colorectal cancer book , Valle et al , but this can be found in many review articles on pubmed also on FAP, for example GeneReviews).

Did the authors include data on all these groups? Than this should be separate graphs.

Screening

This depends on whether or not a patient already had a family history or not. In the first case, usually screening already starts at age 10-12, even when there are no symptoms, leading to an lower age at diagnosis than in symptomatic cases that were not referred for screening.

Methods

Where are the data on FAP and sporadic adenomas/CRC development ages coming from?

In the introduction is mentioned ‘ We selected hereditary and sporadic CRC to investigate because data were available from various population studies on FAP and sporadic patients, no references are given there. Sentence 73, refers to really old and small FAP studies. In these studies was genotype already been taken into account ? Were there no more recent /larger studies available? Can authors contact maybe a clinical FAP research centre for data instead? Reliable Data on ages of CRC in FAP patients are really difficult to get since most of these patients are being screened/operated preventing development of CRC. The question is whether the CRC age data of authors are reliable therefor. Also as stated before, it would be better to look at the number of adenomas developed at different ages for each FAP group separately (severe/classic , moderate/intermediate and mild/attenuated FAP patients). For the general population this I may be more difficult though.

Discussion

In general the argumentation is not always easy to follow.

The question remains what might be an biological explanation why tumor development in the colon is expected to be autocatalytic, except that hits in other tumor suppressor /onco genes follow the initial APC hits. But is this not evolution through accidence in a more dividing cells, or does APC mutations accelerate the occurrence of second hits directly? This is not the case it think since APC does not influence DNA repair. Notable, tumor syndromes in which germline mutation do affect DNA repair, such as Lynch syndrome and Mutyh associated polyposis, an accelerated tumor development has been described. (Evidence for accelerated colorectal adenoma--carcinoma progression in MUTYH-associated polyposis?/ Nieuwenhuis MH et al / Three molecular pathways model colorectal carcinogenesis in Lynch syndrome Ahadova A)

The explanation in part 2 of the big bang theory and how this explain the association of APC and the autocatalytic polymerisation is still not clear to me. The big bang theory explains that the tumor development might not be a linear process but shows acceleration instead, but this is not a proof of APC mutations being essential for this to happen in my opinion. Also not that this non linearity is the same as an autocatalytic reaction.

Part 3: Why is the crypt formation an autocatalytic polymerisation reaction? APC mutations lead to more crypt forming, but how is this accelerated on top of that?. What is the accelerator? Second hits in other genes?

The discussion should be rewritten including more criticism on methods and used data. The conclusions should be less confident and it should be stated clearly that this is still in a very theoretical phase and other approaches are needed to see whether tumor development in the colon indeed is a form of an autocatalyic polymerisation reaction or not.

 Minor:

General: Authors refer to quite old papers in general, thereby not mentioning new developments in more recent papers on Adenoma/CRC development and tumour development in FAP patients

Introduction:

Sentence 40 The scientific community now widely accepts that CRC develops through an adenoma carcinoma sequence [3,4]CRC development can also develop through a CIN, MSI / serrated polyp pathway (without APC mutations but rather BRAF /beta catenin mutations) (see for example Gastroenterology. 2020 Jan;158(2):291-302. doi: 10.1053/j.gastro.2019.08.059. Epub 2019 Oct 14. Pathways of Colorectal Carcinogenesis. Nguyen LH1, Goel A2, Chung DC3.

This is not completely true and this statement should be weakened. Sentence 49 Adenomas in FAP patients show loss of the 2nd wild type APC 49 allele [8-10].APC and the three-hit hypothesis. Segditsas S1, Rowan AJ, Howarth K, Jones A, Leedham S, Wright NA, Gorman P, Chambers W, Domingo E, Roylance RR, Sawyer EJ, Sieber OM, Tomlinson IP.

It should be mentioned here that in patients with attenuated FAP (with germline mutation in the AFAP regions, see before) third APC hits are needed, see the ‘three hit theory’ Sentence 55- 61 Authors should add references for prevalence of FAP (genereviews?) , and adenoma prevalence in the population?

Author Response

Reviewer #2

We thank the reviewers for their helpful and insightful comments, which we believe have significantly improved our manuscript. Our responses to each comment is followed by the arrowhead (â–º).

Comment 1. I really have a problem with the assumption that age at tumor diagnosis can be used as a model for tumor development/progression overall. The age of diagnosis is highly depending on several factors, among which change. Would it not be better to look at the number of adenomas developed at different ages? Or the natural development of polyps/adenomas in one FAP patient?

â–ºWe thank the reviewer for this comment and agree that age of diagnosis can be dependent on several factors. While use of age at tumor diagnosis may not necessarily be a perfect model for tumor development in humans, we believe that “age” still has value for addressing research questions pertaining to mechanisms underlying cancer etiology. Indeed, research on age and cancer risk is a major focus of NCI’s Surveillance, Epidemiology, and End Results program. Previous studies on age at cancer diagnosis have also led to new theories on molecular mechanisms in cancer development such as Knudson’s two hit model. We used age at tumor diagnosis for several reasons: 1) Quality data on relatively large cohorts were readily available including ones on FAP hereditary cancer patients, 2) evaluation of the age at cancer diagnosis provided the initial clue for us to conjecture that cancer development may involve an autocatalytic mechanism. Now that we have put forth our hypothesis on an autocatalytic mechanism, future studies can be designed to test the hypothesis including studies suggested by the reviewer that involve the number of adenomas developed at different ages and the natural development of polyps/adenomas in one FAP patient. We have added this point to future directions in our revised manuscript.

Comment 2 Also assuming that adenoma /CRC incidence at older age slows down (as expected in an autocatalytic reaction) is problematic in my opinion. The question is also whether polyp development ever stops in a FAP patient, as is suggested by authors. I guess we do not know, it seems that even at an old age people keep developing adenomas.

â–ºWe also pondered over this issue but decided to let the data on age at tumor diagnosis speak for itself. In addition, during the course of our study we found other studies that reported that cancer incidence decreases in patients at older ages (85+) for many cancer types [e.g. Cancer in the Oldest Old, Cancer Facts and Figures 2019]. We have now added this point to the discussion.

Comment 3 The less higher incidence of CRC and adenomas at older ages as reported/shown in the graph might be influenced by the fact that in many cases patient have underwent colectomy (in FAP patients) thereby preventing development of CRC . For sporadic cases CRC is being diagnosed less possible not because the incidence goes down but because they are less likely to undergo diagnostic/ colonoscopy because of physical limitations/other diseases.

â–ºWe agree and have added this point to the discussion.

Comment 4 Also, a problem with using the ‘age-at tumor’ diagnosis in FAP is that it is highly influenced by genotype and family history and whether or not people underwent screening already. Differences in phenotype highly relate to the location of the mutation within the APC gene. Three FAP patient groups can be separated: severe/classic, moderate/intermediate and mild/attenuated FAP patients. Did the authors include data on all these groups?

â–ºWe apologize for not being clear about this issue. The available data we analyzed involved a cohort of patients with classical FAP. We have noted this in our revised manuscript including mention of the genotype/phenotype correlation in FAP.

Comment 5 Screening: This depends on whether or not a patient already had a family history or not. In the first case, usually screening already starts at age 10-12, even when there are no symptoms, leading to an lower age at diagnosis than in symptomatic cases that were not referred for screening.

â–ºWe agree and have added this point to the discussion.

Comment 6 Where are the data on FAP and sporadic adenomas/CRC development ages coming from?

â–ºWe apologize for not being clear about this issue. We have clarified this issue in our revised manuscript.

Comment 7 In the introduction is mentioned ‘We selected hereditary and sporadic CRC to investigate because data were available from various population studies on FAP and sporadic patients, no references are given there. Sentence 73, refers to really old and small FAP studies. In these studies was genotype already been taken into account? Were there no more recent /larger studies available? Can authors contact maybe a clinical FAP research centre for data instead?

â–ºWe apologize and have added the references to the sentence noted by the reviewer. We agree that it is important to have large quality population data for analysis. It was for this reason that, at the beginning of our study, we contacted Dr Henry Lynch at Creighton University for recommendations on the best available population studies, particularly since we thought he might have this data in his own hereditary cancer center. It was based on his recommendation that we used the specific data sets that are given in our report.

Comment 8 Reliable Data on ages of CRC in FAP patients are really difficult to get since most of these patients are being screened/operated preventing development of CRC. The question is whether the CRC age data of authors are reliable therefor. Also as stated before, it would be better to look at the number of adenomas developed at different ages for each FAP group separately (severe/classic, moderate/intermediate and mild/attenuated FAP patients). For the general population this I may be more difficult though.

â–ºWe agree. Please see responses to comments 3-5 above.

Comment 9 The question remains what might be an biological explanation why tumor development in the colon is expected to be autocatalytic, except that hits in other tumor suppressor /onco genes follow the initial APC hits. But is this not evolution through accidence in a more dividing cells, or does APC mutations accelerate the occurrence of second hits directly? This is not the case it think since APC does not influence DNA repair. Notable, tumor syndromes in which germline mutation do affect DNA repair, such as Lynch syndrome and Mutyh associated polyposis, an accelerated tumor development has been described. (Evidence for accelerated colorectal adenoma--carcinoma progression in MUTYH-associated polyposis?/ Nieuwenhuis MH et al / Three molecular pathways model colorectal carcinogenesis in Lynch syndrome Ahadova A)

â–ºWe thank the reviewer for raising this important question. We have added some discussion on this point including accelerated tumor development in syndromes in which the germline mutation have an accelerated adenoma-carcinoma progression. We also now discuss genomic instability in colon carcinogenesis from the standpoint of chromosome instability which occurs in ~80% of colon cancers. We also discuss this question in future directions.

Comment 10 The explanation in part 2 of the big bang theory and how this explain the association of APC and the autocatalytic polymerisation is still not clear to me. The big bang theory explains that the tumor development might not be a linear process but shows acceleration instead, but this is not a proof of APC mutations being essential for this to happen in my opinion. Also not that this non linearity is the same as an autocatalytic reaction.

â–ºBased on our participation at national meetings, it appears that research on the big bang theory is still evolving, but what is holding up is that the initiating APC mutation is central to the big bang concept. We thank the reviewer for pointing out the non-linear behavior in the big bang mechanism as well as an autocatalytic mechanism. We have now added this to our paper.

Comment 11 Part 3: Why is the crypt formation an autocatalytic polymerisation reaction? APC mutations lead to more crypt forming, but how is this accelerated on top of that? What is the accelerator? Second hits in other genes?

â–ºWe have worked on the text to clarify that we conjecture that normal crypt renewal and crypt fission are autocatalytic tissue polymerization mechanisms in normal epithelium and that increased autocatalysis occurs due to APC mutation which contributes to colon cancer development. This conjecture concurs with the proliferative shift and increased rate of crypt fission found in APC mutant crypts. What mechanisms might accelerate on top of the effects of APC mutations at this point is unknown and only be speculative at this juncture.

Comment 12 The discussion should be rewritten including more criticism on methods and used data. The conclusions should be less confident and it should be stated clearly that this is still in a very theoretical phase and other approaches are needed to see whether tumor development in the colon indeed is a form of an autocatalyic polymerisation reaction or not.

â–ºWe tried not to oversell our conclusion by stating three times in the manuscript (pages 2,4,6) that our study does not provide proof for an autocatalytic mechanism. Based on the reviewers comment, we now also clearly state in the discussion that the concept is still in a very theoretical phase and other approaches are needed.

Comment 13 General: Authors refer to quite old papers in general, thereby not mentioning new developments in more recent papers on Adenoma/CRC development and tumour development in FAP patients

â–ºWe now mention new developments from more recent studies in the discussion particularly as it relates to comment 4 above.

Comment 14 Introduction: Sentence 40 The scientific community now widely accepts that CRC develops through an adenoma carcinoma sequence [3,4] CRC development can also develop through a CIN, MSI / serrated polyp pathway (without APC mutations but rather BRAF /beta catenin mutations) (see for example Gastroenterology. 2020 Jan;158(2):291-302. doi: 10.1053/j.gastro.2019.08.059. Epub 2019 Oct 14. Pathways of Colorectal Carcinogenesis. Nguyen LH1, Goel A2, Chung DC3). This is not completely true and this statement should be weakened. Sentence 49 Adenomas in FAP patients show loss of the 2nd wild type APC 49 allele [8-10]. APC and the three-hit hypothesis. Segditsas S1, Rowan AJ, Howarth K, Jones A, Leedham S, Wright NA, Gorman P, Chambers W, Domingo E, Roylance RR, Sawyer EJ, Sieber OM, Tomlinson IP. It should be mentioned here that in patients with attenuated FAP (with germline mutation in the AFAP regions, see before) third APC hits are needed, see the ‘three hit theory’ Sentence 55- 61

â–ºThank you for these comments. We have made changes to our revised paper as recommended.

Comment 15 Authors should add references for prevalence of FAP (genereviews?) , and adenoma prevalence in the population?

â–ºThis information has been included with appropriate cited references.

Round 2

Reviewer 2 Report

The authors have included all of my remarks in te text. They did not correct for multiple testing but do not state anymore that the findings are significant.

So at this moment I am satisfied and have no further comments.